# Association of Plasma Vitamins and Carotenoids, DNA Methylation of LCAT, and Risk of Age-Related Macular Degeneration

**DOI:** 10.3390/nu15132985

**Published:** 2023-06-30

**Authors:** Zhaofang Li, Yajing Li, Yijing Hou, Yahui Fan, Hong Jiang, Baoyu Li, Hailu Zhu, Yaning Liu, Lei Zhang, Jie Zhang, Min Wu, Tianyou Ma, Tong Zhao, Le Ma

**Affiliations:** 1School of Public Health, Xi’an Jiaotong University Health Science Center, Xi’an 710061, China; lizhaofang@stu.xjtu.edu.cn (Z.L.); liyajing9611@163.com (Y.L.); houyijing27@163.com (Y.H.); fyh14042166@stu.xjtu.edu.cn (Y.F.); jiangh2015@stu.xjtu.edu.cn (H.J.); 18301644313@163.com (B.L.); zhuhailu@stu.xjtu.edu.cn (H.Z.); lauyaning@163.com (Y.L.); zhanglei701@126.com (L.Z.); zhangjiesxmu@163.com (J.Z.); wumin3779@163.com (M.W.); maty@xjtu.edu.cn (T.M.); 2Key Laboratory of Environment and Genes Related to Diseases, Xi’an Jiaotong University, Ministry of Education of China, Xi’an 710061, China; 3College of Food Engineering and Nutritional Science, Shaanxi Normal University, Xi’an 710119, China; zhaotong@snnu.edu.cn

**Keywords:** age-related macular degeneration, LCAT, DNA methylation, plasma, vitamins and carotenoids

## Abstract

Dysregulation of lipid metabolism has been implicated in age-related macular degeneration (AMD), the leading cause of blindness among the elderly. Lecithin cholesterol acyltransferase (LCAT) is an important enzyme responsible for lipid metabolism, which could be regulated by DNA methylation during the development of various age-related diseases. This study aimed to assess the association between LCAT DNA methylation and the risk of AMD, and to examine whether plasma vitamin and carotenoid concentrations modified this association. A total of 126 cases of AMD and 174 controls were included in the present analysis. LCAT DNA methylation was detected by quantitative real-time methylation-1specific PCR (qMSP). Circulating vitamins and carotenoids were measured using reversed-phase high-performance liquid chromatography (RP-HPLC). DNA methylation of LCAT was significantly higher in patients with AMD than those in the control subjects. After multivariable adjustment, participants in the highest tertile of LCAT DNA methylation had a 5.37-fold higher risk (95% CI: 2.56, 11.28) of AMD compared with those in the lowest tertile. Each standard deviation (SD) increment of LCAT DNA methylation was associated with a 2.23-fold (95% CI: 1.58, 3.13) increased risk of AMD. There was a J-shaped association between LCAT DNA methylation and AMD risk (P_non-linearity_ = 0.03). Higher concentrations of plasma retinol and β-cryptoxanthin were significantly associated with decreased levels of LCAT DNA methylation, with the multivariate-adjusted β coefficient being −0.05 (95% CI: −0.08, −0.01) and −0.25 (95% CI: −0.42, −0.08), respectively. In joint analyses of LCAT DNA methylation and plasma vitamin and carotenoid concentrations, the inverse association between increased LCAT DNA methylation and AMD risk was more pronounced among participants who had a lower concentration of plasma retinol and β-cryptoxanthin. These findings highlight the importance of comprehensively assessing LCAT DNA methylation and increasing vitamin and carotenoid status for the prevention of AMD.

## 1. Introduction

Age-related macular degeneration (AMD) is a leading cause of severe visual impairment among the elderly in the industrialized world. The global prevalence of AMD was estimated to be approximately 8.7% among individuals aged over 45 years, and the number of patients is projected to climb to 288 million in 2040 [1]. The pathological hallmark of AMD is the occurrence of drusen, which consists of extracellular aggregates containing accumulated oxidative stress-derived lipids, suggesting that dysregulation of lipid metabolism contributes to AMD pathogenesis [2]. As the central enzyme for lipid metabolism, Lecithin cholesterol acyltransferase (LCAT) can hydrolyze the oxidation products of lipids, remodel lipoproteins and prevent the accumulation of oxidized derivatives in peripheral tissues [3,4]. Several studies have found that recombinant LCAT or LCAT activators improved abnormal distributions of lipoprotein subfractions and stabilized vulnerable plaques, thereby attenuating atherosclerosis, a disease sharing similar pathological mechanisms with AMD [5,6,7,8]. DNA methylation, the best-characterized epigenetic mechanism, has been reported to repress the transcription of LCAT and be dysregulated during the development of various age-related diseases [9,10]. However, whether LCAT DNA methylation contributes to the pathogenesis of AMD remains elusive.

As efficient nonenzymatic antioxidants in the retina, vitamins and carotenoids were hypothesized to protect against AMD by inhibiting lipid peroxidation through resolving free radical-induced retinal damage along with maintaining an antioxidant reservoir [11,12]. Vitamins and carotenoids may also serve as essential cofactors for DNA methylation enzymes, which regulate the transfer of methyl groups from cytosine to 5-methylcytosine during DNA methylation [13]. Recent studies have reported that several vitamins and carotenoids were inversely associated with DNA methylation of genes involved in lipid metabolism [14,15]. However, studies pertaining to associations between carotenoid and vitamin levels and LCAT DNA methylation are scarce. Moreover, evidence about the association among carotenoid and vitamin concentrations, DNA methylation, and AMD risk has not been well explored.

Therefore, in the present study, we conducted a case–control study among the Chinese population to evaluate the association between DNA methylation levels of the LCAT gene and AMD risk. We also investigated the relationship between circulating vitamin and carotenoid levels and LCAT DNA methylation, as well as their association with AMD risk.

## 2. Materials and Methods

### 2.1. Study Population

Candidates in the present study were participants in the Xi’an Eye Study, a multicenter, double-blind, randomized, and placebo-controlled trial designed to investigate the benefits of lutein and/or fish oil supplementation on the prevention of AMD. Details of the Xi’an Eye Study have been described elsewhere [16]. Briefly, participants were recruited from local communities initiated in 2016. Participants were screened for study eligibility through ophthalmologic examinations, including slit lamp inspection, intraocular pressure, the best-corrected visual acuity, fundus photography, fundus autofluorescence, and optical coherence tomography by ophthalmologists using a standardized protocol. Participants also responded to validated questionnaires inquiring about detailed medical history, lifestyle, and other health information. An overnight blood sample was taken at the same time as the collection of dietary information.

Patients were eligible for inclusion if they were diagnosed with AMD by ophthalmologists in accordance with the Age-Related Eye Disease Study classification system [17]. Cases were excluded from the study if they had high myopia, glaucoma, or cataracts; demonstrated the presence of other macular or choroidal disorders, including diabetic retinopathy, artery occlusion, branch retinal vein, and central serous chorioretinopathy; or had a history of intraocular inflammation, ocular trauma, and prior intraocular surgery within the past 6 months. Those taking medications affecting macular function or who have unstable chronic illnesses were also ineligible. Using the same exclusion criteria, controls were selected from those without clinical signs of AMD. In the current analysis, participants were restricted to those who provided blood samples for testing LCAT DNA methylation and plasma levels of vitamins and carotenes. After exclusions, a total of 126 cases and 174 controls remained in the analysis. 

The study was approved by the Human Research Ethics Committee of the Xi’an Jiaotong University and complied with the tenets of the Declaration of Helsinki. All study subjects gave written informed consent.

### 2.2. Measurement of Circulating Vitamins and Carotenoids

The concentrations of plasma vitamins and carotenoids, including retinol, tocopherol, lutein, carotene, and cryptoxanthin, were determined by reversed-phase high-performance liquid chromatography (RP-HPLC) as previously described [16]. Briefly, a 250 μL plasma sample was precipitated with 250 μL mixing solution (methanol to acetonitrile = 1:2 *v*/*v*) containing 5 μL carotenoid echinenone (CaroteNature, Lupsingen, Switzerland) as an internal standard. The extraction procedure was repeated twice, and the aggregate supernatant was transferred to a vacuum concentration device to evaporate to dryness. After drying the extracts and redissolving them in 100 μL of methanol, 20 μL was injected into the HPLC system. The HPLC system consisted of a YMC-carotenoid C30 analytical column (5 μm, 250 mm × 4.6 mm, YMC Corporation, Kyoto, Japan) and an SPD-M20A diode-array detector (Shimadzu Corporation, Kyoto, Japan). The mobile phase was methanol (mobile phase A) and 50% (*v*/*v*) hexane/isopropanol (mobile phase B). The gradient procedure, at a flow rate of 1 mL/min, was as follows: 0–3 min 0% B, 3–20 min 0–50% B, 20–24 min 50–100% B, 24–30 min 100% B, and 30–31 min 100–0% B. Chromatograms were evaluated at 450 nm for carotenoids and 280 nm for vitamins by using purified external standards. The linearity of the method was established by injecting reference standard in the range of 0.005–50 μg/mL. A correlation coefficient of >0.996 was obtained between the calibration curves of these nutrients.

### 2.3. DNA Methylation Detection of LCAT

The DNA methylation of LCAT was measured by quantitative real-time methylation-specific PCR (qMSP). Genomic DNA was extracted from 200 μL of whole blood using the OMEGA DNA blood mini kit (Omega Bio-tek, Norcross, GA, USA) and bisulfite treated with the EZ DNA Methylation-Gold kit (Zymo Research, Orange, CA, USA), following the manufacturer’s instructions. After sodium bisulfite conversion, LCAT DNA methylation was performed on a LightCycler96 detection system (Bio-Rad, Philadelphia, PA, USA), using SYBR Premix TB Green PCR kit (TaKaRa, Kusatsu, Japan) with the ACTB gene as an internal reference. The relevant qMSP primers were designed using an online MethPrimer [18]. The sequences of the primers were as follows: LCAT (forward: 3′-AGATTGATTAAGATTGAGCGGG-5′ and reverse: 3′-AAATAACTAAAACTACAAATACCCGC-5′) and β-actin (ACTB) (forward: 5′-TGGTGATGGAGGAGGTTTAGAAGT-3′ and reverse: 5′-AACCAATAAAACCTACTCCTCCCTTAA-3′). Real-time PCR was performed in a final reaction volume of 20 μL PCR mixture consisting of 2 μL bisulfite-converted DNA, 10 μL 2XSYBR Premix Ex Taq II (TaKaRa, Kusatsu, Japan), 0.8 μL of each primer, and 6.2 μL DNase-free and RNase-free water. Temperature cycling conditions were optimized as follows: 95 °C for 30 s; 45 cycles of 95 °C for 5 s, 58 °C for 30 s, and 72 °C for 30 s; and a final extension at 72 °C for 5 min. Methylated leukocyte DNA treated with CpG Methylase (M.SssI) was used as a positive control, and untreated DNA was considered as a negative control. Relative LCAT DNA methylation was calculated using the 2^−ΔCt^ method referring to ACTB methylation.

### 2.4. Assessment of Covariates

We obtained information on medical history and lifestyle risk factors for AMD, including age, sex, body weight, educational level, smoking status, alcohol consumption, multivitamin intake, physical activity, personal history of major chronic conditions, and family history of chronic diseases via validated questionnaires. Height and weight were measured using standardized procedures during clinic visits. Alcohol intake was calculated based on the frequency of consumption of beer, wine, and liquor during the previous year. Physical activity was assessed by summarizing hours of moderate to vigorous activity during leisure time per week. Body mass index (BMI) was calculated as weight in kilograms divided by the square of height in meters. Concentrations of plasma lipids, including high-density lipoprotein, low-density lipoprotein, triglycerides, and total cholesterol, were measured using enzymatic methods on an automated biochemistry analyzer (Hitachi 7600-020; Hitachi, Tokyo, Japan).

### 2.5. Statistical Analysis

The participant characteristics were expressed as mean (standard deviation) or median (interquartile range) for continuous variables and percentages for categorical variables. Comparisons of characteristics between cases and controls were performed using Student’s *t*-test or Wilcoxon rank-sum tests for the continuous variables and the chi-square test for the categorical variables. Participants were divided into tertiles according to the distribution of LCAT DNA methylation, whereby the lowest category served as the reference group. A logistic regression model was used to estimate the odds ratio (OR) and 95% confidence interval (CI) for AMD associated with DNA methylation of LCAT. In multivariate analyses, in addition to age and gender, smoking status (yes or no), alcohol intake (yes or no), educational level (less than college or college), physical activity (rarely/never, 1–7 h/w, or >7 h/w), diagnosis of type 2 diabetes (yes or no), and coronary heart disease (yes or no) were adjusted for. To minimize confounding by lipids, we further adjusted for high-density lipoprotein, low-density lipoprotein, triglycerides, and total cholesterol in our final model. Tests for linear trend across the increasing categories of LCAT DNA methylation were performed by assigning the median value of LCAT DNA methylation to the respective categories of exposure and entering this continuous variable into models. Restricted cubic spline regression was used to explore the shape of the association between LCAT DNA methylation and AMD risk, fitting a restricted cubic spline function with four knots (5th, 35th, 65th, and 90th percentiles). In addition, the relationships between carotenoid and vitamin concentrations and LCAT DNA methylation were assessed using linear regression models. For significant results observed in the primary analysis of the relationship between plasma carotenoids and vitamins and LCAT DNA methylation, we further explored the joint associations of these nutrients and LCAT DNA methylation with risk of AMD. The statistical analysis was performed using SPSS for Windows (version 15.0; SPSS). *p* values < 0.05 were considered to indicate statistical significance. 

## 3. Results

The characteristics of AMD cases and controls included in the present study are shown in Table 1. In comparison with the controls, AMD patients were more likely to smoke, consumed more alcohol, were less likely to engage in regular exercise, and tended to have a lower educational status. A history of diabetes or coronary heart disease was also more common among patients with AMD.

Patients with AMD had significantly higher levels of LCAT DNA methylation than the control subjects (Figure 1). In the age- and sex-adjusted model, DNA methylation of LCAT was significantly associated with AMD risk, with the OR comparing extreme categories as 2.88 (95% CI: 1.61, 5.13; P_trend_ < 0.01). The additional control for potential confounding factors somewhat strengthened the association. After adjustment for blood lipids, this association was further strengthened: individuals in the highest tertile of LCAT DNA methylation had a 5.37-fold increased risk of AMD (OR: 5.37, 95% CI: 2.56, 11.28; P_trend_ < 0.01) compared with those in the lowest tertile. Each SD increment of LCAT DNA methylation was associated with a 2.23-fold (95% CI: 1.58, 3.13) increased risk of AMD (Table 2). Spline regression models showed a J-shaped association between LCAT DNA methylation and AMD risk, as the risk of AMD was relatively flat until approximately 1.85 of LCAT DNA methylation and then started to increase rapidly (P_non-linearity_ = 0.03; Figure 2). Using the inflection point of the RCS curve, we performed segmented analysis and found that the OR for a 1 SD increase in LCAT DNA methylation of more than 1.85 was 2.23 (95% CI: 1.21, 2.41; Table 3). 

Table 4 presents the associations between plasma vitamins, carotenoids, and DNA methylation of LCAT. A higher concentration of plasma retinol was significantly associated with a lower level of LCAT DNA methylation (β = −0.05, 95% CI: −0.08, −0.01; *p* < 0.01). A similar inverse association between plasma β-cryptoxanthin and LCAT DNA methylation was observed, with the multivariate-adjusted β of −0.25 (95% CI: −0.42, −0.08, *p* < 0.01). In contrast, weak or null associations with LCAT DNA methylation were observed for the plasma concentrations of other nutrients. 

We further examined the joint associations of plasma vitamins and carotenoids and LCAT DNA methylation with the risk of AMD. Compared with individuals at low LCAT DNA methylation and high plasma retinol, participants who were in the highest tertile of LCAT DNA methylation and the lower level of plasma retinol had the highest risk of AMD, and the OR was 7.01 (95% CI: 2.43, 20.20; Figure 3). A similar pattern for the joint associations was also observed in the combined analyses of β-cryptoxanthin and LCAT DNA methylation, with the highest risk of AMD events occurring in the group with the highest tertile of LCAT DNA methylation and the lower level of plasma β-cryptoxanthin (OR: 3.77, 95% CI: 1.62, 8.80; Figure 4).

## 4. Discussion

In the current study, higher DNA methylation of LCAT was associated with an increased risk of AMD. Plasma vitamins and carotenoids, including retinol and β-cryptoxanthin, were significantly inversely associated with DNA methylation of LCAT. The inverse association between increased LCAT DNA methylation and AMD risk was more pronounced among participants who had a lower concentration of plasma retinol and β-cryptoxanthin.

Oxidation of lipids accumulated in the retina leads to the formation of drusen, a characteristic lesion of AMD, indicating that dysregulation of lipid metabolism is involved in the pathogenesis of AMD [19,20,21]. As a central enzyme involved in lipid metabolism, LCAT is able to hydrolyze oxidized lipids and prevent the oxidation of lipoproteins, which are associated with age-related diseases such as AMD and atherosclerosis. In a cross-sectional study among 105 participants with LCAT deficiency disorders, heterozygotes for LCAT gene defects, who present with compromised LCAT function, exhibit an increased risk for atherosclerosis [22]. Raphaël et al. also observed that decreased LCAT function, as a result of LCAT gene mutations with 31% lower LCAT activity levels, is associated with increased atherosclerotic plaque components [23]. Atherosclerosis shares a similar pathogenesis to AMD, with atherosclerotic plaques and ocular drusen both being over-accumulation of extracellular deposits and having the same molecular components [24,25]. The present study, so far as we know, is the first to examine the association between DNA methylation of LCAT and AMD risk and finds that higher LCAT DNA methylation was associated with an increased risk of AMD. Potential mechanisms through which LCAT hypermethylation increases AMD risk include impairing lipid efflux, inducing oxidative stress, and accelerating inflammation. Elevated levels of LCAT DNA methylation can physically prevent transcription factor binding to a regulatory region of DNA, which is associated with the gene silencing of LCAT, subsequently leading to the dysregulation of lipid efflux [9,26,27]. Defective lipid efflux has been shown to increase lipid accumulation and subsequently generate inflammatory monocytes that infiltrate the retina, ultimately causing photoreceptor dysfunction and neurodegeneration [28,29]. A study in apoA-I^-/-^ mice demonstrated that LCAT would modulate the expression of apoA-IV, which can promote the de novo biogenesis of discrete HDL-A-IV particles that promote lipid efflux [30]. In previous in vitro experiments, Czarnecka et al. found that LCAT activity could hamper the exchange of free cholesterol from HDL to macrophages, thereby facilitating the net cellular clearance of lipids [31]. LCAT hypermethylation might also impair LCAT activity and consequently increase oxidative damage to lipids and ultimately be implicated in the generation of atherosclerosis, a disease that shares similar pathologic mechanisms with AMD. Through inducing low-density lipoprotein oxidation and impairing high-density lipoprotein antioxidant defense, LCAT deficiency is associated with increased oxidative stress and higher macrophage homing and leads to atherosclerotic lesions in dyslipidemic obese mice models [32]. In previous in vitro experiments, Vohl et al. revealed that LCAT can prevent the oxidation of both phospholipids and cholesteryl esters by utilizing a catalytically active serine residue as a proton donor in vitro [33]. In addition, LCAT contains Sp binding sites and interacts with specificity protein one, which is able to induce pro-inflammatory cytokine expression via activating the Wnt/β-catenin signaling pathway, which has been implicated in the degeneration of the focal retina and exudative lesions [34,35,36,37].

Previous studies have found that higher levels of vitamins and carotenoids were associated with an increase in the activity of LCAT. In a randomized controlled trial, McEneny et al. found that lycopene-rich diets, either as tomato-rich foods or supplements, could enhance LCAT-HDL3 activities after a 12-week intervention among 225 healthy adults [38]. Daniels et al. showed that a carotenoid enrichment in HDL contributed to the increased activity of LCAT among 80 patients with diabetes with a high vegetable and fruit diet [39]. The LCAT activity may be modified by the level of LCAT DNA methylation, and the present study showed that higher circulating concentrations of retinol and β-cryptoxanthin were associated with a lower DNA methylation of LCAT. Compared with other vitamins and carotenoids, retinol and β-cryptoxanthin revealed higher receptor binding affinity for retinoic acid receptors (RARs) [40,41]. By binding to and transactivating RARs, these nutrients are responsible for the formation of the visual pigment rhodopsin, which exhibits photoprotective effects and restores visual function in retinal degeneration [42,43]. RARs have been shown to recruit epigenetic enzymes to specific target gene promoters and interact with other epigenetic proteins, such as the methyl-CpG-binding protein MBD1, and the Polycomb repressive complex 2, resulting in CpG island hypermethylation [44]. Nonetheless, more mechanistic studies are needed to elucidate the relationships between circulating retinol and β-cryptoxanthin levels and LCAT DNA methylation.

It should be noted that the detrimental association of LCAT hypermethylation with AMD risks is exacerbated by low concentrations of retinol and β-cryptoxanthin. One possible mechanism underlying the divergent associations on AMD by nutrient status is perhaps that upon low nutrient levels, LCAT became hypermethylated, possibly due to increased oxidative stress. Decreased concentrations of retinol and β-cryptoxanthin may result in higher oxidative stress status, causing enhanced lipid peroxidation products to accumulate in the retina and leading to amorphous deposits, which contribute to the progressive degeneration of photoreceptors and loss of central vision in AMD [2]. Through inducing cellular oxidative stress, O’Hagan et al. reported that hydrogen peroxide treatment recruits DNA methyltransferases (DNMTs), along with SIRT1, to move from non-GC-rich regions into CGIs, where two subunits of the PRC2 complex are recruited, resulting in an increase in DNA methylation [45]. Another mechanism is probably pertinent to the higher expression level of pro-inflammatory cytokines, and thus the adverse effects of LCAT hypermethylation are stronger among participants with low nutrient levels. Elevated pro-inflammatory cytokine levels have been found in patients suffering from vitamin deficiency [46,47,48]. Evidence has been accumulated that pro-inflammatory cytokines, including IL-1β and IL-6, induced inflammasomes and increased the susceptibility of RPE cells to cytotoxicity in retinal degeneration [49]. Activation of IL-6 induces increased chromatin binding of epigenetic proteins, including DNMT1, EZH2, and SIRT1, later manifesting in aberrant DNA hypermethylation [50]. LACT hypermethylation may also modulate the bioavailability of retinol and β-cryptoxanthin by a negative feedback regulation, which further supports the key role of these compounds in the relationship between hypermethylated LCAT and AMD risk. Using 2-DE proteomic of plasma samples from 38 carriers of LCAT mutations, Simonelli et al. observed that LCAT deficiency syndromes caused by mutations in the LCAT gene exhibit an increased level of retinol-binding protein 4, which is the sole specific transport protein for retinol [51]. Unlike retinol, the transport of β-cryptoxanthin requires spherical HDLs, the maturation of HDLs that are converted from discoidal HDLs and modulated by LCAT. In adenovirus-mediated gene transfer of apoA-I (apolipoprotein A-I) mutants in the apoA-I^-/-^ mouse model, Koukos et al. reported that the defects of mutants that inhibit the conversion of discoidal to spherical HDL could be corrected by adenovirus overexpressing LCAT [52]. In the present study, the deleterious associations of LACT hypermethylation with AMD risks could be aggravated among participants with lower carotenoid and vitamin levels, emphasizing the value of increasing the circulating carotenoids and vitamins on retina health.

Several limitations should be considered in interpreting the results. First, because of the observational nature of the present study, it is impossible to establish a cause–effect relationship between LCAT DNA methylation and the risk of AMD. Similar to other observational studies, residual confounding remains a possibility. However, several important established and potential risk factors for AMD have been carefully adjusted in the current analysis, which may minimize residual confounding to a certain extent. Second, LCAT DNA methylation was measured in peripheral tissues rather than in the primary affected tissues, such as the retina in AMD. Although retinal tissues are ideal for studying the disease, it remains challenging to obtain retinas from living patients. Studies have attempted to compare methylation patterns among retina and peripheral tissues and found a strong correlation of methylation profiles between tissues, with a similar proportion of CpG sites being methylated [53,54,55]. Therefore, measuring the methylation level of LCAT in blood still provides useful insights into the pathogenesis of AMD. Third, measurements of vitamins and carotenoids were only measured at a single point in time, which may not have closely represented the long-term circulating range. However, among the New York University Women’s Health Study, reasonable validity has been demonstrated with intraclass correlation coefficients (interclass correlation coefficients ranged between 0.63 and 0.85) over a three-year period, indicating that a single measurement of vitamin and carotenoid concentrations could provide a reliable estimate of long-term levels over time [56]. Fourth, increased DNA methylations are thought to be commonly related to the down-regulation of gene expression [56]. Previous studies have reported a moderate to strong inverse relationship between LCAT DNA methylation and LCAT mRNA expression [27,57,58]. These previous results were in agreement with our comparative study of LCAT mRNA expression in human donor retina both with and without AMD, which found that the retinal level of LCAT mRNA expression was significantly lower in eyes with AMD than non-AMD eyes (Please see Appendix A). Further studies may also be needed to better elucidate the biological mechanism underlying this association. Lastly, because our participants comprised residents with Chinese ancestry, the generalizability of our findings to other groups may be limited.

## 5. Conclusions

In summary, the results of this study indicated that LCAT hypermethylation was associated with an increased risk of AMD. The increased risk is more prominent for participants with low concentrations of vitamins and carotenoids. These findings highlight the importance of comprehensively assessing LCAT DNA methylation and increasing vitamin and carotenoid status in AMD prevention. Future prospective studies to investigate the underlying mechanisms of LCAT hypermethylation to AMD risk over different statuses of vitamins and carotenoids are needed.

## Figures and Tables

**Figure 1 nutrients-15-02985-f001:**
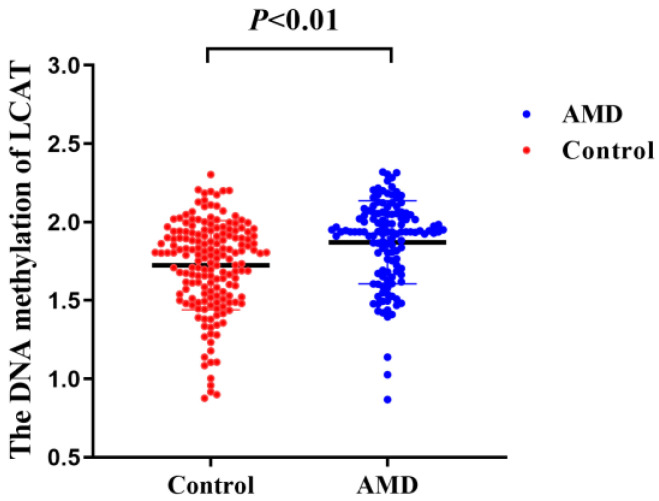
The DNA methylation of LCAT is increased in AMD patients compared with the control subjects. AMD, age-related macular degeneration; LCAT, lecithin-cholesterol acyltransferase.

**Figure 2 nutrients-15-02985-f002:**
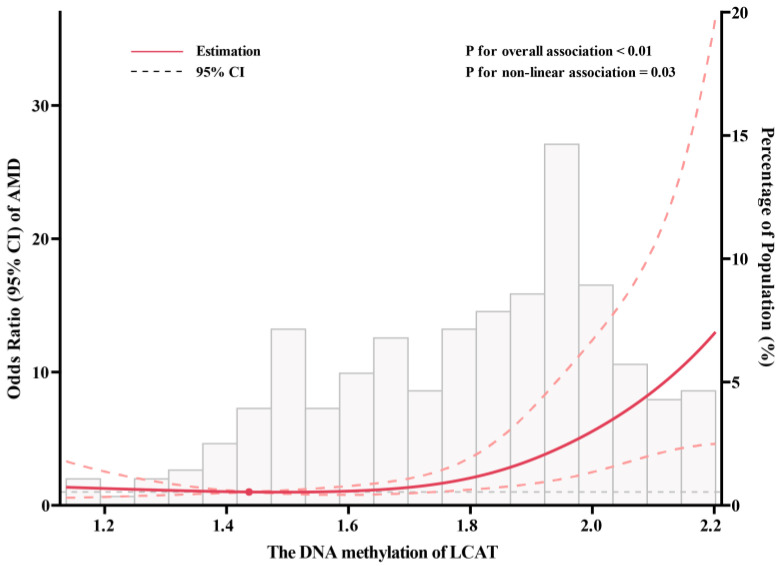
Restricted cubic spline analysis of association between DNA methylation of LCAT and AMD risks. Adjusted for age (continuous), sex (men or women), cigarette smoking (never/ever), alcohol drinking (never/ever), education level (less than college or college), physical activity (rarely/never, 1–7 h/w, or >7 h/w), diagnosis of type 2 diabetes (yes or no) and diagnosis of coronary heart disease (yes or no), high-density lipoprotein (continuous), low-density lipoprotein (continuous), triglycerides (continuous), and total cholesterol (continuous). Solid line is point estimate, and dashed lines are 95% confidence intervals. AMD, age-related macular degeneration; LCAT, lecithin-cholesterol acyltransferase. Solid line is point estimate, and dashed lines are 95% CIs.

**Figure 3 nutrients-15-02985-f003:**
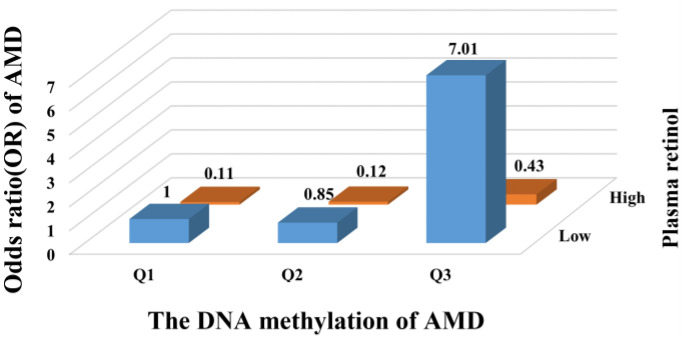
Joint association of LCAT DNA methylation and plasma retinol with risk of AMD. Odds ratio was calculated in logistic models after adjusting for age (continuous), sex (men or women), cigarette smoking (never/ever), alcohol drinking (never/ever), education level (less than college or college), physical activity (rarely/never, 1–7 h/w, or >7 h/w), diagnosis of type 2 diabetes (yes or no) and diagnosis of coronary heart disease (yes or no), high-density lipoprotein (continuous), low-density lipoprotein (continuous), triglycerides (continuous), and total cholesterol (continuous). AMD, age-related macular degeneration; LCAT, lecithin-cholesterol acyltransferase.

**Figure 4 nutrients-15-02985-f004:**
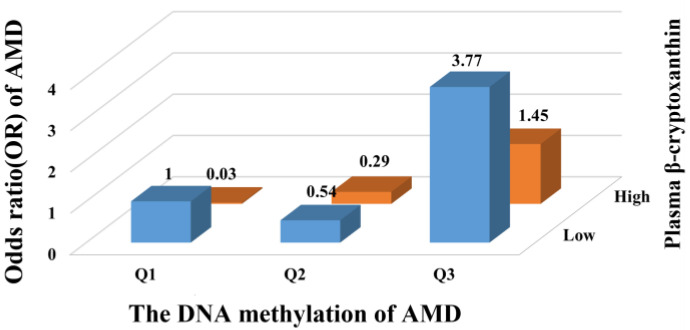
Joint association of LCAT DNA methylation and plasma β-cryptoxanthin with risk of AMD. Odds ratio was calculated in logistic models after adjusting for age (continuous), sex (men or women), cigarette smoking (never/ever), alcohol drinking (never/ever), education level (less than college or college), physical activity (rarely/never, 1–7 h/w, or >7 h/w), diagnosis of type 2 diabetes (yes or no) and diagnosis of coronary heart disease (yes or no), high-density lipoprotein (continuous), low-density lipoprotein (continuous), triglycerides (continuous), and total cholesterol (continuous). AMD, age-related macular degeneration; LCAT, lecithin-cholesterol acyltransferase.

**Table 1 nutrients-15-02985-t001:** Baseline characteristics of the study population.

Variables	Controls (*N* = 174)	AMD Patients (*N* = 126)	*p*
Age, years	65.45 ± 7.60	66.99 ± 7.37	0.08
Male, %	37.93	40.47	0.66
Cigarette smoking, %	10.91	29.37	<0.01
Alcohol consumption, %	7.47	15.08	0.04
Education Level, %			<0.01
Less than college	36.21	61.90	
College	63.79	38.10	
Physical activity, %			<0.01
>7 h/w	31.03	13.50	
1–7 h/w	50.57	35.71	
Rarely/never	18.39	50.79	
Multivitamin use, %	21.26	27.78	0.19
BMI, kg/m^2^	23.42 ± 1.43	23.69 ± 1.29	0.10
History of diabetes, %	10.92	21.43	0.01
History of coronary heart disease, %	6.90	15.87	0.01
Family history of diabetes, %	20.11	19.05	0.82
Family history of coronary heart disease, %	27.59	20.63	0.17

Abbreviations: AMD, age-related macular degeneration. Categorical variables are expressed as n (%), and the continuous variables are expressed as the mean SD values.

**Table 2 nutrients-15-02985-t002:** Adjusted odds ratios (95% confidence interval) for AMD according to tertiles of LCAT DNA methylation.

DNA Methylation	Odds Ratios (95% CIs) by Tertiles of LCAT DNA Methylation			LCAT Increase, per 1 SD	*P_trend_*
	T1	T2	T3		
DNA methylation level, median	1.48	1.79	1.99		
No. of cases/controls	26/58	24/58	76/58		
Model 1	1	0.93 (0.48, 1.81)	2.88 (1.61, 5.13)	1.75 (1.34, 2.29)	<0.01
Model 2	1	0.87 (0.39, 1.96)	5.09 (2.46, 10.53)	2.18 (1.56, 3.06)	<0.01
Model 3	1	0.91 (0.40, 2.05)	5.37 (2.56, 11.28)	2.23 (1.58, 3.13)	<0.01

Abbreviations: OR, odds ratio; CI, confidence interval; AMD, age-related macular degeneration; LCAT, Lecithin-cholesterol acyltransferase; T1, the lowest tertile of LCAT DNA methylation; T2, the middle tertile of LCAT DNA methylation; and T3, the highest tertile of LCAT DNA methylation. Model 1: adjusted for age and sex. Model 2: additionally adjusted for cigarette smoking(never/ever), alcohol drinking (never/ever), education level (less than college or college), physical activity (rarely/never, 1–7 h/w, or >7 h/w), diagnosis of type 2 diabetes (yes or no), and diagnosis of coronary heart disease (yes or no). Model 3: additionally adjusted for high-density lipoprotein (continuous), low-density lipoprotein (continuous), triglycerides (continuous), and total cholesterol (continuous).

**Table 3 nutrients-15-02985-t003:** Adjusted odds ratios (95% confidence interval) for AMD according to LCAT DNA methylation from segmented logistic regression analysis.

Characteristic	Model 1		Model 2		Model 3	
	OR (95% CI)	*p*	OR (95% CI)	*p*	OR (95% CI)	*p*
LCAT DNA methylation increase per 1 SD						
<1.85	1.14 (0.83, 1.59)	0.42	1.05 (0.74, 1.52)	0.80	1.13 (0.78, 1.67)	0.54
≥1.85	1.19 (0.81, 1.79)	0.39	1.47 (1.05, 2.12)	0.03	1.68 (1.21, 2.41)	<0.01

Abbreviations: OR, odds ratio; CI, confidence interval; AMD, age-related macular degeneration; LCAT, Lecithin-cholesterol acyltransferase. Model 1: adjusted for age and sex. Model 2: additionally adjusted for cigarette smoking(never/ever), alcohol drinking (never/ever), education level (less than college or college), physical activity (rarely/never, 1–7 h/w, or >7 h/w), diagnosis of type 2 diabetes (yes or no), and diagnosis of coronary heart disease (yes or no). Model 3: additionally adjusted for high-density lipoprotein (continuous), low-density lipoprotein (continuous), triglycerides (continuous), and total cholesterol (continuous).

**Table 4 nutrients-15-02985-t004:** Association between vitamin and carotenoid concentrations and LCAT DNA methylation.

Vitamins and Carotenoids	β	95% CI	SE	*p*
Retinol (µmol/L)	−0.05	(−0.08, −0.01)	0.02	<0.01
α-tocopherol (µmol/L)	−0.24	(−0.49, 0.01)	0.13	0.06
Lutein (µmol/L)	0.04	(−0.09, 0.17)	0.07	0.53
β-cryptoxanthin (µmol/L)	−0.25	(−0.42, −0.08)	0.09	<0.01
β-carotene (µmol/L)	−0.25	(−0.78, 0.26)	0.27	0.35
Lycopene (µmol/L)	0.13	(−0.05, 0.32)	0.09	0.16

Abbreviations: β, coefficients; CI, confidence interval; SE, standard error. Adjusted for age (continuous), sex (men or women), cigarette smoking (never/ever), alcohol drinking (never/ever), education level (less than college or college), physical activity (rarely/never, 1–7 h/w, or >7 h/w), diagnosis of type 2 diabetes (yes or no) and diagnosis of coronary heart disease (yes or no), high-density lipoprotein (continuous), low-density lipoprotein (continuous), triglycerides (continuous), and total cholesterol (continuous).

## Data Availability

The original contributions presented in this study are included in the article; further inquiries can be directed to the corresponding authors.

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
