# Peer review of "Association of Plasma Vitamins and Carotenoids, DNA Methylation of LCAT, and Risk of Age-Related Macular Degeneration"

_nutrients, 2023, doi:10.3390/nu15132985_

Round 1
Reviewer 1 Report
This manuscript describes measurement of dna methylation of plasma and its relation to age-related macular degeneration, vitamins and carotenoids.
They have measured the concentrations of these substances in the plasma, and found that lcat methylation has a significant correlation with amd.
Furthermore, they found that plasma retinol and beta-cryptoxanthin has a negative relationship with dna methylation.
These findings are very critical for the public health and is worth acceptance in Nutrients.
Some minor errors should be addressed.
P3 assessment of covariates
Concrete data and calculation method should be described.
P6 table 2 explanation on q1, q2 and q3 should be added.
P9 L25 from bottom LACT LCAT
no problem
Reviewer 2 Report
The manuscript by Li at al. provides an interesting study on the correlation between DNA methylation of lecithin cholesterol acyltransferase, an enzyme important in lipid metabolism and the risk of developing an age-related macular degeneration. Also, the potential involvement of several plasma vitamins and carotenoids in this association is assessed as well.
The manuscript is very well written. The methodology is well set and methods are described in detail. Lots of consideration is given to selection of candidates. The strongest part of the manuscript is the discussion. The authors have discussed their results in regard to known findings in the subject. And at the end, all of the potential limitations of the presented work are mentioned and also shortly explained.
In Material and methods, line 99, I presume it is an error stating 250 L (litres) of plasma?
I would only suggest in the tables to highlight the significant changes (bold or different colour).
Line 44. Omit “:” after Lecithin
In table 3, correct “methylation”
Reviewer 3 Report
The authors investigated the relation between LCAT DNA methylation and risk of age-related macular degeneration, in this study. The design of this study is well-organized and the reviewer will accept this manuscript if the authors respond to some points below.
Major
To what extent is the level of the protein or mRNA expression related to the LCAT DNA methylation? It is better for adding the data of LCAT mRNA or protein expression in the patients.
Minor
1. Throughout the whole manuscript, “LCAT methylation” may not be a common and clear expression. It may be better as “DNA methylation of LCAT” or “LCAT DNA methylation” as the authors used in the title and other places.
2. Line 44: Lecithin: cholesterol acyltransferase --> Lecithin cholesterol acyltransferase
Round 2
Reviewer 3 Report
The authors responded to the reviewer's comment clearly. It is worth publishing. I'm glad to read your research.